# Parenting, Gender, and Perception of Changes in Children’s Behavior during the COVID-19 Pandemic

**DOI:** 10.3390/ijerph20156452

**Published:** 2023-07-27

**Authors:** Jael Vargas-Rubilar, María Cristina Richaud, Cinthia Balabanian, Viviana Lemos

**Affiliations:** 1National Council for Scientific and Technical Research (CONICET), Buenos Aires C1425FQD, Argentina; 2Centro Interdisciplinario de Investigaciones en Ciencias de la Salud y del Comportamiento (CIICSAC), Universidad Adventista del Plata, Libertador San Martín 3103, Argentina; 3Instituto de Ciencias de la Familia (ICF), Universidad Austral, Buenos Aires B1630FHB, Argentina

**Keywords:** parenting, sex, behavioral changes, pandemic

## Abstract

In a previous Argentine study, we found that, in the critical context of social isolation during the COVID-19 pandemic, there were changes in maternal practices that influenced the relationship with their children. We also found that the impact of mandatory isolation was moderated positively by protective factors such as positive parenting and maternal school support or negatively by risk factors such as maternal stress. Although this study only analyzed maternal behavior, we were interested in studying the behavior of both parents, comparing the parenting (positive parenting, parental stress, and school support) of the father and mother and the perceived behavioral changes in their children. A quantitative ex post facto study was carried out. The sample consisted of 120 Argentinean parents (70 mothers and 50 fathers) aged between 27 and 56 (*M* = 38.84; *SD* = 5.03). Questionnaires were administered on sociodemographic and behavioral data of the children, as well as a brief scale to assess parenting. Mann–Whitney U and MANOVA were used to analyze the influence of gender on perceived changes in children’s behavior and perceived parenting, respectively. Mothers perceived more significant changes than fathers in their children’s behavior. In addition, women reported more parental stress, greater child school support, and greater perceived positive parenting compared to men. These findings support the hypothesis that parenting developed differently in fathers and mothers. These results imply the need for psycho-educational intervention programs aimed at promoting greater involvement of fathers in parenting and better management of parental stress in mothers’ and family psychological well-being.

## 1. Introduction

The COVID-19 pandemic was an extraordinary and unprecedented phenomenon because of its global impact. Combating the disease and developing ways to prevent it in a short time was one of the greatest challenges the world has faced in recent decades. Consequently, many of the measures taken by governments to contain the virus (e.g., compulsory social isolation) had a negative impact on the mental health of family groups. In this direction, previous studies, e.g., ref. [1] had already pointed out that social isolation during pandemics or natural disasters can generate psychological disorders (e.g., post-traumatic stress) for both parents and children. Indeed, initial studies of people’s reactions to the pandemic context revealed symptoms of anxiety and depression, as well as symptoms of stress in the population [2,3]. In addition, the rapid spread of this virus provoked fear of the disease, as well as fear of the social and economic consequences of the pandemic [4]. Particularly, in Argentina, a study conducted on the general population [5] showed that anxiety and depressive symptoms increased over time and that intolerance to uncertainty was the main predictor of this variability. Especially women and young people reported the highest number of psycho-pathological symptoms [5].

During the COVID-19 pandemic, compulsory social isolation, social and work conditions, and changes in education negatively influenced parenting practices, e.g., [6,7,8]. Many parents faced work changes such as job loss, reduced pay, and remote work, and at the same time, had to assume greater childcare responsibilities due to school closures [9]. In this direction, several studies have reported that many parents felt overloaded and stressed by having to perform full-time parenting [6,10]. In addition, parents or caregivers reported higher alcohol consumption, more irritability, lower positive communication, and more mental health problems [3,8,11]. Likewise, the COVID-19 pandemic harmed women with children more [12], and this effect was greater in mothers caring for more children [13]. Mothers or caregivers suffered a greater load of unpaid work and the requirement of multifunctionality in relation to having to fulfill work, domestic, and child care roles, especially in Latin American cultures [14], such as Argentina.

The literature in the area has pointed out that positive parenting can act as a protective factor for children against stressful events [15] and as a facilitator of family resilience [16]. Similarly, an Argentine study [13] analyzed maternal perceptions of three dimensions of parenting (i.e., positive parenting, parental stress, and school support) and how these impacted perceived behavioral changes in children (e.g., in sleep, appetite, mood, obedience, fighting with siblings, participation and attitude in online classes, etc.). The results showed that, indeed, the type of parenting influenced the perceived behavior of the children during the pandemic. Women with low positive parenting reported that children were more disobedient, fought and yelled more, had more defiant and dependent behaviors, and were more nervous/anxious. Mothers with higher parental stress perceived more negative changes in most of their children’s behaviors. In addition, they reported that their children showed more sadness and regressive behaviors in relation to the less stressed mothers. Women who reported having provided more school support also perceived that children adapted better to online classes, doing their homework, and enjoying their classes while being less frustrated by having to do schoolwork at home [13].

As for the children, a study that analyzed the impact of quarantine on Italian and Spanish children and adolescents [17] showed that more than 80% of parents perceived negative changes in their mood and behavior during social isolation. The most frequent changes observed in children were: difficulty in concentrating, boredom, irritability, restlessness, nervousness, feelings of loneliness, discomfort, and fears. Particularly in Argentina, a study conducted during the pandemic by COVID-19 [18] in 4500 children and adolescents found that almost 8 out of 10 (77%) showed more anger and 68% felt sad, 7 out of 10 children and adolescents (6 to 18 years of age) expressed feelings of discouragement and boredom, and 60% reported fear. Sixty percent felt a lack of outdoor recreational activities and sports, especially children and adolescents aged 6 to 14 years [18].

In recent decades, we have witnessed important changes in favor of gender equality that have resulted in greater involvement of fathers in the care and education of children. Nonetheless, parenting remains the most gender-typed role during adulthood [19,20]. Mothers around the world show greater availability and commitment than fathers to parenting. Some studies conducted in families in Kenya, India, Guatemala, and Peru revealed that fathers rarely participate in the care of children under one year of age [21]. In this direction, a recent study conducted in the United States [22] indicated wide differences in the way mothers and fathers describe parenting styles. For example, about half of mothers (51%) say they tend to be overprotective compared to 38% of fathers. In turn, fathers (24%) are more likely to give children too much freedom than mothers (16%). Mothers are also more likely than fathers to say that parenting is exhausting (47% vs. 34%) and stressful (33% vs. 24%) all or most of the time [22]. Consistent with previous surveys [23], mothers report taking more responsibility for children’s care than fathers, although fathers tend to say they share responsibilities almost equally. Most mothers (78%) report assuming more responsibility for managing their children’s schedules and supervising homework than their husbands (65% of women with school-age children), providing emotional support and attention to their sons and daughters (58%), and satisfying the basic needs of their children: care, hygiene, and food (57% with children under five years of age). Therefore, it can be stated that women already assumed a greater share of children’s care before the pandemic. Even when both parents work, women have struggled more with work-family balance [24].

Specifically, during the pandemic, women reported a greater load of child care and housework [14] and more symptoms of anxiety and exhaustion, as well as more parenting-related worries and fears than men [9]. However, a Canadian study suggests that fathers increased their involvement in household and children-care tasks during this period [25].

In this context, the present study aimed to compare changes in children’s behaviors and different parenting aspects as perceived by fathers and mothers during the COVID-19 pandemic.

Based on the proposed objectives, the following hypotheses were formulated:

**Hypothesis** **1.**
*There are differences between fathers and mothers regarding the perception of behavioral changes in children during the pandemic.*


**Hypothesis** **2.**
*Mothers perceive higher levels of parental stress, involvement in children’s school support, and positive parenting compared to fathers during the pandemic.*


## 2. Materials and Methods

### 2.1. Type of Study and Design

The study was quantitative based on empirical data, descriptive, ex post facto, as it involves a comparison of DV across groups classified according to categorized and assigned variables [26]. In addition, it is cross-sectional since the study data were collected at a single point in time [27].

### 2.2. Participants

The sample consisted of 120 parents (70 mothers, 58%, and 50 fathers, 42%) who accessed the online form and answered it completely. Participants were parents of school children aged between 27 and 56 years (*M* = 38.84; *SD* = 5.03) from different Argentine provinces. A non-probabilistic availability sampling method was used [28]. In addition, we asked about the work status of the fathers and mothers in the sample and found that 57% were employed, 31% were self-employed, and the remaining 12% were unemployed or homemakers who performed domestic and care tasks in their homes, without working outside the home. Finally, when evaluating the educational background of the parents, it was found that half of them had a university degree, and 5% had postgraduate studies (see Table 1).

The inclusion criteria used were: (a) being over 18 years of age and (b) being a parent of schoolchildren (+5 and 12 years of age). Parents or caregivers of children with psychological, developmental, or learning disorders were not included in the study.

### 2.3. Instruments

Sociodemographic questionnaire. To collect information on the sociodemographic characteristics of the participants, an ad hoc semi-structured questionnaire was used to collect data on gender, age, occupation, and educational level.

Questionnaire of perceived behavior of children. A semi-structured questionnaire was designed to assess changes in children’s behavior as perceived by parents during the pandemic. In the questionnaire, parents were asked whether they observed variations in their children’s behavior. The questionnaire mainly inquired about (a) behaviors observed in the children concerning sleep, eating habits, mood, and relationship with siblings and friends, and (b) behaviors observed concerning the children’s school situation: online classes, relationship with classmates, and compliance with school assignments. Response options were: less, the same, or more than before the pandemic (e.g., eating: less, the same, more). The content validity of the instrument was studied based on the criteria of expert judges, who were asked whether the behaviors assessed were relevant to analyze in the context of a pandemic. An adequate Aiken’s V coefficient (between 0.8 and 1) was obtained [29].

The brief scale of perceived parenting during the pandemic [30]. This instrument assesses three dimensions of perceived parenting in the pandemic context: (a) positive parenting (e.g., I dedicate some time during the day to speak to my children), (b) parenting stress (e.g., Time is not enough, as it used to be, to fulfill all my responsibilities), and (c) parenting school support (e.g., I know which homework and assignments are given to my children in online education), based on 17 items with a 4-point Likert-type response scale (i.e., Never, Seldom, Very often, and Always). The study of the instrument carried out in Argentina indicated adequate psychometric properties [30]. The confirmatory factor analysis of three factors of the scale showed satisfactory fit indexes (χ^2^/gL = 1.22; NFI = 0.93; NNFI = 0.99; CFI = 0.99; IFI = 0.99; GFI = 0.99) and an acceptable error (RMSEA = 0.02). The reliability (i.e., internal consistency) was acceptable for the three dimensions: positive parenting (ω = 0.79), parenting stress (ω = 0.77), and school support (ω = 0.75) [30].

### 2.4. Ethical Procedures and Data Collection

All actions performed in the setting of this study followed the international ethical recommendations for research involving human subjects [31]. In all cases, the purpose of the research was explained, and parents gave informed consent before completing the form. Responses were anonymous, and data confidentiality was strictly guarded. A reduced number of questions was included to avoid participant fatigue. The response time was less than 10 min.

Due to the confinement conditions of the pandemic, the invitation to participate in the study was made through social networks (Facebook 289.0.0.20.120. and Instagram 161.0.0.41.122), email (Gmail 2020.08.23.335177159, Outlook 4.2038.2, etc.), and instant messaging services (WhatsApp 2.20.197.21 and Telegram 7.0.1, etc.). Data collection was conducted during the social isolation phase between September and December 2020. The information was collected through an online form (i.e., Google Forms 2.20.301.07.46), which included the instruments described in the previous section.

### 2.5. Procedures for Data Analysis

For the description of the sociodemographic variables and the study variables, descriptive statistics were calculated: measures of central tendency (mean, standard deviations), frequencies, and percentages. In addition, the skewness and kurtosis of the dimensions of perceived parenting were calculated. A Multivariate Analysis of Variance (MANOVA) was performed to compare perceived parenting as a function of parental gender.

Given the qualitative nature of the variables, the Mann–Whitney U test was used to analyze changes in behavior as a function of parental gender.

Data analysis was performed with SPSS version 25 software.

## 3. Results

### 3.1. Perception of Changes in Children’s Behavior between Fathers and Mothers

Table 2 shows which children’s behaviors were perceived as different between fathers and mothers during the pandemic. In the behaviors: is anxious/nervous, screams, and has nightmares, mothers perceived significantly more changes than fathers. For the remaining behaviors assessed, mothers also perceived more changes in their children than fathers, although these differences were not statistically significant (see Table 2). These results partially support Hypothesis 1.

### 3.2. Comparison of Perceived Parenting between Fathers and Mothers

The skewness and kurtosis values of the dependent variables analyzed were below ±1, as recommended for parametric analyses [32,33]. As shown in Table 3, the dimensions of parenting were differentially perceived between fathers and mothers (Hotelling’s F(3, 116) *p* < 0.001; η^2^ = 0.23). Both positive parenting (F(1, 118) = 7.16; *p* = 0.009; η^2^ = 0.06), parental stress (F(1, 118) = 19.09; *p* < 0.001; η^2^ = 0.14), and involvement in school support (F(1, 118) = 18.44; *p* < 0.001; η^2^ = 0.14) presented higher scores for mothers than for fathers. (See Table 3). These findings support Hypothesis 2.

## 4. Discussion

The COVID-19 pandemic has brought numerous challenges worldwide and had a strong impact on interpersonal relationships in general and particularly on family dynamics, routines, and interactions. Also, mothers and fathers often perceive and exercise their family roles, parenting styles, and practices differently [34]. Consequently, the present study aimed to compare changes in children’s behaviors and parenting as perceived by fathers and mothers during the COVID-19 pandemic in Argentina.

Overall, the results of this study indicated differences between fathers and mothers in perceived changes in some behaviors of their children and perceived parenting (i.e., positive parenting, parental stress, and school support) during the pandemic.

Regarding behavioral changes in children during the pandemic, differences were observed in the perception of fathers and mothers. Mothers, in particular, perceived more changes in all the behaviors evaluated. We hypothesize that since, in our culture, the mother is usually the most influential figure in parenting [35] and is more aware of the health, school performance, and interpersonal relationships of children, she has developed a greater perception of changes in children’s behavior in adverse situations. At the same time, although mothers perceived changes to a greater extent in all the behaviors assessed, they did so especially in: shows dependent behavior, is anxious/nervous, screams, and has nightmares. In this sense, the child behaviors in which the greatest differences were observed in comparison to those perceived by the fathers were those referring to the children’s emotional problems. Indeed, some studies suggest that mothers tend to perceive their children’s emotional changes more easily and to be more sensitive to the signals their children give compared to fathers [36,37]. Regarding nightmares, one study indicated that mothers were more likely to report sleep problems in their children [38]. We initially thought that differences in the perception of behavioral changes would be significant for all behaviors. However, it is possible that, during social isolation, parents may have taken a more active role in children’s care compared to before the pandemic. Some studies have suggested that fathers increased their involvement in performing household and children care tasks during the pandemic, e.g., [25]. The compulsory social isolation, which lasted for several months in Argentina, led many parents to work remotely and thus spend more time at home and possibly interact more with their children. Thus, the results obtained partially support our first hypothesis: There are differences between fathers and mothers regarding the perception of behavioral changes in children during the pandemic.

On the other hand, the results showed significant differences in perceived parenting between fathers and mothers. First, mothers perceived that they had more positive parenting practices than fathers during the pandemic. These results are consistent with pre-pandemic studies, so this aspect would seem to respond more to socio-cultural influences than to the pandemic context. For example, in a study conducted in the Netherlands, mothers reported significantly more positive parenting practices than fathers [39]. In another pre-pandemic study in the Mexican population, mothers reported more authoritative parenting strategies than fathers, which was confirmed by their partners [40]. Also, a study conducted in Chinese families [34] found differences in perceived parenting between fathers and mothers. Mothers reported having a more authoritative (i.e., warm and less controlling) parenting style than fathers. In other works, e.g., [41,42], mothers scored on average higher than fathers on all positive parenting practices

Regarding parental stress, it appeared more highlighted in mothers, possibly because they were mainly responsible for children’s care and housework [43], while many of them had to continue with their work. In this direction, a US study by [44] showed that 79% of mothers reported being primarily responsible for housework and 66% for children’s care during the pandemic, compared to 28% and 24% of fathers, respectively. Likewise, findings from a study conducted in the United Kingdom and Ireland [45] showed that female caregivers reduced time spent at work and significantly increased time spent caring for children compared to men during the pandemic. Another study of Norwegian mothers [46] indicated that their well-being decreased significantly compared to before confinement. Furthermore, it was found that gender ideologies aggravated the negative impact of increased domestic responsibilities (i.e., children care and housework) on the well-being (higher level of stress) of mothers. Mothers who more strongly endorsed the belief that mothers are instinctively and innately better caregivers than fathers, increasing their perceptions of increased household responsibilities, perceived lower well-being after confinement [46]. In this direction, a study [47] that analyzed parenting practices during the pandemic in five cultures (i.e., Bulgaria, Israeli Arabs, and Israeli Jews, Spain, and the United States) found some similarities across cultures. Particularly, collaborative behaviors in the home were less common among men in all cultures assessed.

Finally, mothers reported providing significantly more school support to their children than fathers. In this direction, some studies [44] have pointed out that the division of time devoted to learning at home and monitoring homework completion during the pandemic was also influenced by gender. A total of 84% of mothers reported spending more time providing school support than other family members, compared to 50% of fathers. Mothers were also more likely (57%) to say that they felt more pressure regarding their children’s home learning compared to fathers (45%). Indeed, this study showed that of couples working from home, 72% of mothers perceived themselves to be primarily responsible for children’s care compared to 33% of men. In this sense, mothers may have felt more responsible for their children’s education even when both parents were available at home for this task. Similarly, another study conducted during the pandemic [48] found that mothers were 10 times more likely to be in charge of their children’s education than fathers, and 44% of mothers felt that they had no help with education at home and, for this reason, reported being more stressed than fathers.

The results for the three parenting aspects that indicated higher values for mothers support our second hypothesis: Mothers perceive higher levels of parental stress, involvement in children’s school support, and positive parenting compared to fathers during the pandemic.

## 5. Limitations and Strengths

The main strength of this study is that it allows us to know how a challenging context such as the COVID-19 pandemic affected some characteristics of parenting. In addition, it provides insight into the perceptions of fathers and mothers concerning parenting in Argentine culture.

However, the study conducted has some limitations that should be considered. First, the sample was purposive and non-probabilistic, and its size was small, making it unrepresentative of the Argentine population. For example, families from other social strata were not represented in these results, which is an important limitation considering that families of low socioeconomic status and with pre-existing problems in family relationships or mental health were more affected by multiple collateral effects of the pandemic [49,50].

In addition, due to the conditions of social isolation, self-reports of the parents were used, and the children’s perspective could not be considered in the evaluation. Future studies should also assess parenting and behavioral changes perceived by the children themselves. Regarding the evaluation of behavioral changes, only content validity was studied. The stability of the instrument could not be studied due to the fluctuating conditions of the pandemic context. On the other hand, since this was a cross-sectional study, it is not possible to determine whether the changes in behavior and perceived parenting were maintained over time or were modified after the pandemic. Therefore, it would be advisable to conduct a post-pandemic evaluation to know the long-term effects of the pandemic on both caregivers and children. Also, in this paper, we have mainly analyzed the role of gender in parenting and the perception of children’s behavior. However, there are many other contextual variables and individual differences that could be analyzed in future studies, such as social status, gender of children, age of parents and children, occupation, and number of children, among others.

## 6. Implications

The results of the present study allow for delineating intervention strategies in challenging contexts (e.g., pandemics or natural disasters) in the future. At the same time, it highlights the need to design, implement and evaluate parenting intervention programs to mitigate the long-term negative effects of the COVID-19 pandemic on parenting. These approaches should consider the differences in parenting beliefs and practices that exist between fathers and mothers. Such proposals could promote greater involvement of men in parenting. In particular, they should provide psycho-educational training to parents or caregivers to contain children and prevent their psychological distress. Indeed, the protective role that caregivers can have in the face of fear and stress during a pandemic is important [51], so timely intervention is necessary to ensure the mental health of all family members. Intervention programs should be based on sound theoretical approaches such as positive parenting, which has had promising results in adverse contexts and situations, e.g., [35,52,53,54]. Likewise, strategies for the management of parental stress, especially in mothers, should be included in the program, considering the damage it can have on the family atmosphere, parenting, and psychological well-being of children, e.g., [55].

## 7. Conclusions

In short, during the COVID-19 pandemic quarantine, mothers perceived more behavioral changes than fathers. They also reported more positive parenting, more parental stress, and more school support than fathers. In this sense, we could say that a risk context such as social isolation during the COVID-19 pandemic highlighted and exposed characteristics that are present in the usual conditions of families and that, therefore, are more determined by already existing social patterns, in this case referring to gender differences present in the different cultures according to the studies reviewed. These findings make visible the specific challenges faced by families, especially mothers, during the critical period of compulsory social isolation during the pandemic. In particular, they highlight the importance of addressing the gender differences that imposed additional loads on women in families.

## Figures and Tables

**Table 1 ijerph-20-06452-t001:** Sociodemographic characteristics of the participants.

	*N*	%
Gender		
Female	70	58.3%
Male	50	41.7%
Civil status		
Single	5	4.2%
In consensual union	23	19.2%
Married	75	62.5%
Separated	5	4.2%
Divorced	9	7.5%
Other	3	2.5%
Occupation		
Homemaker	14	11.7%
Unemployed	1	0.8%
Self-employed	37	30.8%
Employed	68	56.7%
Academic studies		
Primary school	1	0.8%
Secondary school	27	22.5%
Tertiary education (non-university level)	26	21.7%
University degree	60	50%
Postgraduate studies (incomplete)	1	0.8%
Postgraduate studies (complete)	5	4.2%

**Table 2 ijerph-20-06452-t002:** Range difference in perception of children’s behavioral changes between fathers and mothers.

	Mother	Father	U	*p*
Is sad	64.06	55.52	1501	0.155
Disobeys	60.7	60.22	1736	0.936
Fights with siblings	64.36	55.09	1479.5	0.117
Is anxious/nervous	66.66	51.87	1318.5	0.015
Screams	67.84	50.22	1236	0.003
Shows dependent behavior	65.26	53.84	1417	0.053
Shows defiant behavior	62.99	57.01	1575.5	0.326
Once asleep, he/she wakes up confused in the middle of the night	63.74	55.97	1523.5	0.182
Has nightmares	65.64	53.3	1390	0.040
Gets easily frustrated when doing school assignments	64.06	55.52	1501	0.157

**Table 3 ijerph-20-06452-t003:** Comparison of perceived parenting between fathers and mothers.

Dimensions of Parenting	Mothers		Fathers		F	*p*
	*M*	*DE*	*M*	*DE*		
Positive parenting	3.38	0.40	3.18	0.45	7.16	0.009
Parenting stress	2.52	0.54	2.05	0.64	19.09	0.000
School support	3.30	0.71	2.72	0.76	18.44	0.000

## Data Availability

The raw data supporting the conclusions of this article will be made available by the corresponding author. The data are not publicly available due to privacy or ethical restrictions.

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
