# Peer review of "Parenting, Gender, and Perception of Changes in Children’s Behavior during the COVID-19 Pandemic"

_ijerph, 2023, doi:10.3390/ijerph20156452_

Round 1
Reviewer 1 Report
The manuscript “Parenting, Gender, and Perception of Changes in Children's Behavior During the COVID-19 Pandemic” present an Argentine study aimed at better understanding the behavior of both parents, comparing the parenting (positive parenting, parental stress, and school support) of the father and mother and the perceived behavioral changes in their children during the Covid-19 lockdown. The study is well written and the topic is interesting. The methodology is correct and I appreciated the statistical plan and analyses. I would enrich the discussion with comparing authors’ results with those derived from the international literature, as there is a lot of evidence about parenting during the pandemic. The severity of pandemic preventive measures varied per countries, just as the outcomes may have varied, so I would add a paragraph about cross cultural comparisons.
Minor revisions
Eliminate in line 24 “is likely influenced by culture”as you did not investigate the role of culture
I will skip the following lines: 131-134 This section may be divided by subheadings. It should provide a concise and precise description of the experimental results, their interpretation, as well as the experimental conclusions that can be drawn.
Line 187 correct in anxious/nervous,
An ethical committee approval is lacking
Eliminate Figure 1 that is redundant with Table 1
The same for Figure 2
In Table 2 add the third decimal in the p column. You cannot write p= .00
Author Response
Responses to the reviewer are detailed below the editor's comments.
1.The study is well written and the topic is interesting. The methodology is correct and I appreciated the statistical plan and analyses. I would enrich the discussion with comparing authors' results with those derived from the international literature, as there is a lot of evidence about parenting during the pandemic. The severity of pandemic preventive measures varied by countries, just as the outcomes may have varied, so I would add a paragraph about cross cultural comparisons.
We appreciate all the suggestions made by the reviewer to optimize our paper.
We include some references from international and cross-cultural studies. In line with our study, in all of them we found that women took more care of parenting during the pandemic and we understand that this responds to a socio-cultural pattern. The pandemic reinforced what is already established in most cultures.
2. Eliminate in line 24 “is likely influenced by culture” as you did not investigate the role of culture
Done! Thanks!
3. I will skip the following lines: 131-134 This section may be divided by subheadings. It should provide a concise and precise description of the experimental results, their interpretation, as well as the experimental conclusions that can be drawn.
Done. We removed the sentence suggested (from line 131-134).
4. Line 187 correct in anxious/nervous.
Done. Changed text to anxious/nervous.
5. An ethical committee approval is lacking.
The ethics’ committee resolution was added.
6. Eliminate Figure 1 that is redundant with Table 1. The same for Figure 2
Figures 1 and 2 have been removed from the text.
7. In Table 2 add the third decimal in the p column. You cannot write p= .00
We added the third decimal in the Table. Thanks for the suggestion!
Reviewer 2 Report
1. ABTRACT
Please include what is the implication of this study
2.1 Type of study and Design
Too Short please elaborate what type of quantitative design is this and justify why this design been apply for this study.
2.2 Participants
What is the population of this study. Why only 120 samples. How did you identify this sample?
Data collection via social media. Are there any sampling errors or all the 120 selected samples were completed the questionnaires.
2.3 Instruments
The instruments were designed by the researchers. Have you conducted pilot study and conduct reliability and validity of these questionnaires.
2.5 Procedures for data analysis
Have you conducted the normality test. What is the result?
3. Results
Please include descriptive analysis of the respondents
Just use table 1 or figure 1 to analyze the result.
3.1 Perception of changes in children's behavior between fathers and mothers
What is the finding? accepting or rejecting the HYPOTHESES 1
Table 1: Explain how the u and p value calculated.
3.2 Comparison of perceived parenting between fathers and mothers
What is the finding? accepting or rejecting the HYPOTHESES 2
OVERALL
Please add more research question and hypothesis.
Author Response
We appreciate all the suggestions made by the reviewer to optimize our paper.
Responses are detailed below each reviewer's comment.
1. ABSTRACT
Please include what is the implication of this study
Done. The implications were included in the abstract.
2. Type of study and Design: Too short please elaborate what type of quantitative design is this and justify why this design has been applied for this study.
Done. We detailed the type of quantitative design used and the corresponding justification for its use in this study.
3. Are there any sampling errors or all the 120 selected samples were completed the questionnaires?
The online form used (Google form) only allows for the questionnaires that are fully answered to be counted.
4. Instruments. The instruments were designed by the researchers. Have you conducted pilot study and conduct reliability and validity of these questionnaires?
Regarding the children's behavior questionnaire, its reliability cannot be examined (test-retest) because it was assessed during the pandemic. This limitation was added in the manuscript. The psychometric properties of the parenting scale are already described in the instruments section.
5. Procedures for data analysis. Have you conducted the normality test? What is the result?
Possible biases with respect to normality were analyzed through the asymmetry and kurtosis of the variables involved in the study (see table in attachment document). These figures did not exceed the values of ±1 recommended for parametric analysis (Muthén & Kaplan, 1992).
6. Results. Please include descriptive analysis of the respondents
Done. Table 1 was included in the Participants section with the descriptive analyses.
7. Just use table 1 or figure 1 to analyze the result.
Done. We removed Figure 1.
8. Perception of changes in children's behavior between fathers and mothers. What is the finding? accepting or rejecting the HYPOTHESES 1
Done This was added in Results, but it is also developed in the Discussion.
9. Table 1: Explain how the u and p value calculated.
Mann-Whitney U is applied when the assumptions required for parametric tests cannot be made. The software used to calculate the Mann-Whitney U and p was SPSS version 25, non-parametric test chart.
10. Comparison of perceived parenting between fathers and mothers. What is the finding? accepting or rejecting the HYPOTHESES 2 1
Done. Added in Results, but also developed in Discussion
11. Please add more research question and hypothesis.
The suggestion is not clear. The objective and the hypotheses of the study are explicit. Does it refer to the proposal of new questions and hypotheses based on the results obtained? New research questions are suggested for future studies, they are developed in the last paragraph of the Strengths and Limitations section.

Reviewer 3 Report
Dear authors and editor,
The manuscript titled "Parenting, Gender, and Perception of Changes in Children's 2 Behavior During the COVID-19 Pandemic" aimed to compare changes in children's behaviors and different parenting aspects as perceived by fathers and mothers during the COVID- 19 pandemic.
There are many minor and major issues I'd like the authors resolve.
Abstract
1-Add the study design to the abstract. Also, the authors can choose to add the study design to the title of the manuscript.
2-Change the keywords. Not found in the MeSH (Medical Subject Headings): gender; children’s behavioral changes
Introduction
3-The introduction is adequate as it sets out the background to the topic and summarizes the key concepts. In addition to correctly state the objectives of the study.
Materials and Methods
4-It is also recommended to add ethical permissions.
5-Study size: Explain how the study size was arrived at. The sample size is very important.
Results
6-Put the same decimal places in the p values.
Discussion
7-It is recommended to start the discussion by bringing together the most relevant results.
8-Adequate, the authors include the most important findings and limitations of the study. Add as a limitation the need for further psychometric analysis of the scale.
Conclusions
9-Adequate: Answer the objectives clearly and concisely.
References
10-Adequate
Author Response
We appreciate all the suggestions given to optimize the presentation of our paper.
Responses are detailed below each reviewer's comment.
Abstract
1-Add the study design to the abstract. Also, the authors can choose to add the study design to the title of the manuscript.
We appreciate the reviewer's suggestions. We found it more appropriate to add the design in the abstract and not in the title which was already long.
2- Change the keywords. Not found in the MeSH (Medical Subject Headings): gender; children's behavioral changes Done.
Changed keywords: sex to gender, and behavior changes to children's behavioral changes. Thanks!
Introduction
3-The introduction is adequate as it sets out the background to the topic and summarizes the key concepts. In addition to correctly state the objectives of the study.
Materials and Methods
4-It is also recommended to add ethical permissions.
Done. We added the reference to the approval of the ethics committee (p. 9, line 356) and sent the informed consent to the publisher.
5-Study size: Explain how the study size was arrived at The sample size is very important.
All fathers and mothers who answered the entire form during the period of social isolation during the pandemic (between September and early December) were included in the study. Data collection was not continued after that date because the conditions of social isolation changed significantly in Argentina.
Results
6- Put the same decimal places in the p values.
Done. Thanks!
Discussion
7- It is recommended to start the discussion by bringing together the most relevant results.
Done! See in line 218-220.
8-Adequate, the authors include the most important findings and limitations of the study. Add as a limitation the need for further psychometric analyses of the scale
Done.The impossibility of studying the stability of the instrument (test- retest) of the questionnaire due to the fluctuating conditions of the pandemic context was included as a limitation. The psychometric properties of the parenting scale were reported in the Instruments section.
Round 2
Reviewer 2 Report
Please check your references. some are missing.